# In Vitro Evaluation of Essential Oils and Saturated Fatty Acids for Repellency against the Old-World Sand Fly, *Phlebotomus papatasi* (Scopoli) (Diptera: Psychodidae)

**DOI:** 10.3390/insects15030155

**Published:** 2024-02-24

**Authors:** Kevin B. Temeyer, Kristie G. Schlechte, Joel R. Coats, Charles L. Cantrell, Rodrigo Rosario-Cruz, Kimberly H. Lohmeyer, Adalberto A. Pérez de León, Andrew Y. Li

**Affiliations:** 1Knippling-Bushland U. S. Livestock Insects Research Laboratory and Veterinary Pest Genomics Center, USDA-ARS, 2700 Fredericksburg Road, Kerrville, TX 78028, USA; kevin.temeyer@usda.gov (K.B.T.); kristie.schlechte@usda.gov (K.G.S.); kim.lohmeyer@usda.gov (K.H.L.); beto.perezdeleon@usda.gov (A.A.P.d.L.); 2Pesticide Toxicology Laboratory, Department of Plant Pathology, Entomology and Microbiology, 2007 ATRB, Pammell Drive, Iowa State University, Ames, IA 50011, USA; jcoats@iastate.edu; 3Natural Products Utilization Research Unit, USDA-ARS, Thad Cochran Research Center, University, MS 38677, USA; charles.cantrell@usda.gov; 4BioSA Research Lab., Natural Sciences College, Campus el ‘Shalako’ Las Petaquillas, Autonomous Guerrero State University, Chilpancingo 62105, Guerrero, Mexico; rockdrig@yahoo.com.mx

**Keywords:** sand flies, *Phlebotomus papatasi*, repellents, DEET, IR3535, essential oils, fatty acids

## Abstract

**Simple Summary:**

The old-world sand fly, *Phlebotomus papatasi* (Scopoli 1786), is a major vector of *Leishmania major*, the predominant pathogen responsible for zoonotic cutaneous leishmaniasis in the Middle East, North Africa, Southern Europe, and Central Asia. DEET and other synthetic insect repellents have been used for personal protection against sand fly bites. However, the frequent use of DEET repellent raised concerns in regards to skin sensitivity, toxicity, and unpleasant odor. There are increasing efforts to evaluate natural products for use in developing more effective organic sand fly repellents. This paper reports the results of a laboratory study on several plant essential oils and saturated fatty acids concerning their repellency against female sand flies. A static air repellency assay was used to measure the responses of sand flies to test materials. The sand fly repellency of each test material was compared with those achieved by commercial repellent DEET and IR35353 at the same test concentration. The study identified two of the tested essential oils as effective spatial repellents at reduced concentrations compared to those of DEET, and two saturated fatty acids were found to produce significant sand fly mortality. The results from this study establish the foundation for developing more effective natural sand fly repellent products.

**Abstract:**

The sand fly, *Phlebotomus papatasi* (Scopoli, 1786), is a major vector for Leishmania major in the Middle East, which has impacted human health and US military operations in the area, demonstrating the need to develop effective sand fly control and repellent options. Here, we report the results of spatial repellency and avoidance experiments in a static air olfactometer using the female *P. papatasi* testing essential oils of *Lippia graveolens* (Mexican oregano), *Pimenta dioica* (allspice), *Amyris balsamifera* (amyris), *Nepeta cataria* (catnip), *Mentha piperita* (peppermint), and *Melaleuca alternifolia* (tea tree); the 9–12 carbon saturated fatty acids (nonanoic acid, decanoic acid, undecanoic acid, and dodecanoic acid); and the synthetic repellents DEET and IR3535. The materials applied at 1% exhibited varying activity levels but were not significantly different in mean repellency and avoidance from DEET and IR3535, except in regards to nonanoic acid. Some materials, particularly nonanoic and undecanoic acids, produced sand fly mortality. The observed trends in mean repellency over exposure time included the following: (1) *P. dioica* oil, M. alternifolia oil, decanoic acid, undecanoic acid, DEET, and IR3535 exhibited increasing mean repellency over time; (2) oils of *N. cataria*, *A. balsamifera*, *M. piperita*, and dodecanoic acid exhibited relatively constant mean repellency over time; and (3) *L. graveolens* oil and nonanoic acid exhibited a general decrease in mean repellent activity over time. These studies identified the essential oils of *N. cataria* and *A. balsamifera* as effective spatial repellents at reduced concentrations compared to those of DEET. Additional research is required to elucidate the modes of action and potential synergism of repellents and essential oil components for enhanced repellency activity.

## 1. Introduction

Sand flies are vectors of intracellular protozoa and other pathogens causing leishmaniasis and other diseases affecting humans and animals predominantly in tropical and semi-tropical areas of the world [1,2]. The old-world sand fly, *Phlebotomus papatasi* (Scopoli 1786), is a major vector of *Leishmania major*, the principal pathogen responsible for zoonotic cutaneous leishmaniasis in the Middle East, Asia, Africa, and Southern Europe [3,4,5,6,7]. Adult female *P. papatasi* transmit *L. major* when they bite humans and animals to feed on blood, which is necessary for reproduction [1].

Old-world sand flies have significantly impacted US military operations and readiness in Iraq and Afghanistan [8,9,10]. Control efforts involving the application of chemical insecticides at U.S. military bases were ineffective, and sand fly bites resulted in significantly reduced availability of military personnel to conduct daily operations [11,12,13]. The repellent *N,N*-diethyl-m-toluamide (DEET) was effective for short-term use [7,14,15]. However, the need for routine reapplication of DEET was inconvenient for deployed personnel, and this repellent did not clear or protect tents and other human habitations from sand fly encroachment and concealment. The frequent use of DEET also raised concerns in regards to skin sensitivity, toxicity, and unpleasant odor [16,17]. Effective repellent technologies for personal application, domicile protection, and area-wide use were needed [9]. This situation prompted research to identify alternative sand fly control technologies, including repellents.

Natural products such as plant essential oils and fatty acids are known to exhibit repellent or insecticidal activity against hematophagous arthropod disease vectors [18,19,20]. Essential oils are generally regarded as safe relative to synthetic compounds with insecticidal or repellent activities [21,22,23,24,25]. Although the insecticidal and repellent effects of essential oils or fatty acids against mosquitoes and ticks were documented repeatedly [26,27], reports of similar investigations concerning *P. papatasi* and other old-world sand fly species are limited [14,28,29]. Yaghoobi-Ershadi et al. [30] reported that the essential oil of myrtle, *Myrtus communis*, and DEET were effective repellents to *P. papatasi*; however, DEET exhibited higher repellent efficacy. Kimutai et al. [31] reported that the essential oils of lemon grass, *Cymbopogon citratus*, and Mexican marigold, *Targetes minuta*, were effective repellents and biting deterrents, protecting treated hamsters from starved *P. duboscqi*, an important vector of *L. major* in Eastern Africa, for up to 3 h, but DEET was effective at significantly lower doses. In Kenya, plant extracts from *Tarchonanthus camphoratus*, *Acalypha fruticose*, and *Tagetes minuta* exhibited repellent and insecticidal activity against *P. duboscqi* [32,33].

Investigations on the traditional use of plants by indigenous populations to repel biting insects revealed that fatty acids contributed to the repellency effect [34,35]. These studies documented biting deterrence to the mosquito *Aedes aegypti* using K&D module and cloth patch bioassay systems and established fatty acid activity comparable, or superior, to DEET. Decanoic, undecanoic, and dodecanoic acids in breadfruit (*Artocarpus altilis*) were the primary deterrent constituents and were significantly more effective than DEET [35]. Subsequently, the biting deterrent effects of a series of saturated fatty acids against *Ae. aegypti* were also determined [18,36,37]. Medium-chain fatty acids (C 10:0 to C 13:0) demonstrated the highest biting deterrence, followed by acids with short (C 6:0 to C 9:0), and greater chain lengths (C 14:0 to C 18:0), reflecting a trend of repellent activity by fatty acids against diverse arthropod pests and disease vectors. These observations led us to hypothesize that fatty acids might also be repellent to sand flies.

Laboratory studies on the repellency of natural products against sand flies need to include DEET because this synthetic compound is considered the gold standard, enabling comparison of repellent activity among different studies [7,29,38]. DEET appears to act by targeting octopaminergic synapses, producing neuroexcitation and toxicity to insects [39].

Ethyl butylacetylaminopropionate, or EBAAP, also known as IR3535, is another synthetic insect repellent structurally similar to β-alanine or pantothenic acid, which is widely used in Europe [40,41,42]. IR3535 repels and deters biting by *P. duboscqi*, *P. mascittii*, and *P. papatasi* [7,43]. IR3535 activates insect muscarinic M1 acetylcholine receptors, inducing an increase in intracellular calcium concentration [41].

Terpenes are common constituents of plant essential oils, and some act through inhibition of the enzymatic activity of acetylcholinesterase (AChE) [44]. Inhibition of AChE activity has been reported for many different essential oils [45,46,47,48,49,50,51]. In addition, DEET is reported to reduce expression of AChE in the tick, *Dermacentor variabilis* [52,53], which can be easily being confused with AChE inhibition. Our search of the available literature failed to locate reports of tests of IR3535 for inhibition of AChE in sand flies. Biochemically active recombinant AChE of *Phlebotomus paptasi* (rPpAChE1) was readily available in our laboratory [54,55], providing the opportunity to test in vitro whether IR3535 and DEET inhibited the recombinant AChE of *P. papatasi*.

The primary objective of this study was to evaluate the repellent activities of various essential oils and several 9–12 carbon saturated fatty acids against adult sand flies in comparison to two commercial repellent compounds, DEET and IR3535.

## 2. Materials and Methods

### 2.1. Sand Flies

A colony of *P. papatasi*, strain Israeli, was established at the U.S. Department of Agriculture-Agricultural Research Service (USDA-ARS) Knippling-Bushland U.S. Livestock Insects Research Laboratory (KBUSLIRL) in 2010 using sand flies from a colony at the Walter Reed Army Institute of Research (WRAIR, Silver Spring, MD), maintained since 1983 according to mass rearing procedures [56,57]. Briefly, ~300 adult, blood-fed *P. papatasi* were collected weekly by manual aspiration, anesthetized under CO_2_, transferred into Nalgene^®^ 500 mL clear plastic larval pots (egging chambers; holes drilled in the bottom, with a water-saturated plaster base and screw tops with screens for air access), and incubated at 26 °C, 85% humidity, for 1 week to allow for egg deposition. After oviposition, the dead adults were removed by aspiration, larval medium was added, and incubation continued, with periodic larval feeding, for an additional 1–2 weeks to allow for maturation and pupation. Larval chambers were monitored daily for adult fly emergence. Newly emerged adult flies were released into adult plexiglass cages (Figure 1A), and the estimated enumeration of released flies was recorded. Adult flies were fed daily with cotton pads saturated with 30% sucrose and twice each week with 37 °C defibrinated bovine blood using mosquito feeding tubes (25 mm, #34-17280-25, Kimble/Chase Custom Glass Shop, Vineland, NJ, USA) covered with porcine intestinal membrane (sausage casing) obtained from a local slaughterhouse (Figure 1B,C). The KBUSLIRL sand fly colony was maintained for 9 years and used for research in support of the Deployed Warfare Fighter Protection Program, jointly administered by the Armed Forces Pest Management Board (U.S. Department of Defense) and the USDA-ARS. The KBUSLIRL colony generally produced over 10,000 adult flies per week, as previously described ([58,59,60] Figure 1).

### 2.2. Essential Oils and Chemical Compounds

The essential oils of Mexican oregano (*Lippia graveolens*) and allspice (*Pimenta dioica*) were obtained by steam distillation, as previously described [61,62]. Essential oil of amyris (*Amyris balsamifera*) was prepared as described by Paluch et al. [16,63]. Essential oil of catnip (*Nepeta cataria*) was prepared as described previously [64,65]. Essential oils of *Mentha piperita* (triple-rectified peppermint oil) and *Melaleuca alternifolia* (century tea tree oil) were obtained from Jindal Drugs Ltd. and 21st Century Healthcare, respectively, by Joel Coats. Reagent grade nonanoic acid, decanoic acid, dodecanoic acid, and DEET were purchased from Sigma-Aldrich (St. Louis, MO, USA). Undecanoic acid was purchased from Fluka (Sigma-Aldrich, St. Louis, MO, USA). IR3535 was the generous gift of Merck KgaA (Darmstadt, Germany).

### 2.3. Static Air Repellency Bioassays

Repellency bioassays were conducted in a static-air chamber at 25–26 °C, as described by Paluch et al. [16,63]. All test materials were tested and compared at a concentration of 1% diluted in ethanol. DEET, IR3535, and several top essential oils were further tested and compared at lower concentrations (0.5%, 0.25%, 0.125%, 0.0625%, 0.03125%). For each bioassay, twenty adult female *P. papatasi* sand flies (3–7 days post emergence, not blood-fed) were collected by manual aspiration, anesthetized with CO_2_, and placed in the static-air chamber. Opposite ends of the static air chamber were closed by plastic Petri plate lids containing 9 mm (63.6 cm^2^) Whatman No. 1 filter discs treated with 1 mL of acetone (control) or test material diluted in acetone and allowed to air dry on aluminum foil for 1 h. The static air chamber was placed on top of a marked paper, dividing the chamber into 4 quadrants (Figure 1D,E), and numbers of flies on the treated or control paper surfaces was recorded at 15, 30, 60, 90, 120, and 180 min. Additionally, the number of flies in each quadrant of the chamber was also recorded at 15, 30, 60, 90, 120, and 180 min, with flies on the filter papers at the ends of the chamber included in the number of flies for the appropriate end quadrant. DEET (97% pure, technical grade, Sigma-Aldrich, St. Louis, MO, USA) was used as a reference standard for comparison of relative repellent or avoidant activity [29]). Repellency (spatial repellency), avoidance (contact repellency), and insecticidal activities were calculated for each time point, as described by Paluch et al. [16,64]. Each repellency experiment was performed in triplicate, while the bioassays for the solvent present at both ends of the tube included five replicates.

### 2.4. Inhibition of P. papatasi Acetylcholinesterase

Biochemically active recombinant acetylcholinesterase of *P. papatasi* (rPpAChE1) was expressed in the baculoviral system, as previously described [54]. Nonanoic acid, undecanoic acid, IR3535, and DEET were tested for inhibition of rPpAChE1 activity, essentially as described in Temeyer et al. [55]. Recombinant PpAChE1 was preincubated for 15 min. with test concentrations of the inhibitor prior to initiation of the reaction by the addition of a substrate. Initial reaction velocities at each inhibitor concentration were used to determine inhibition compared to that of the uninhibited enzyme.

### 2.5. Data Analysis

Repellency (spatial) and avoidance (contact) were calculated for each of the six time points during the course of the bioassay using the following formulas [16,63]:Repellency (%) = [(# flies in untreated half chamber − # flies in treatedhalf)/(total # flies)] × 100%
Avoidance (%) = [(# flies on untreated paper − # flies on treated paper)/(#flies on untreated paper + # flies on treated paper)] × 100%

The repellency value for a replicate was generated by averaging the repellency values of all six time points. The avoidance value for a replicate was generated by averaging the avoidance values of all six time points. The accumulative fly mortality (%) for a replicate was determined using this formula:Mortality (%) = [(# dead flies [total over the 180 min])/(total # flies)] × 100%

One-way ANOVA was used to compare the effects of test materials in regards to their mean repellency, avoidance, and mortality using JMP v. 12 software (SAS Institute Inc., Cary, NC, USA). The Tukey–Kramer HSD Test, using the same software, was employed for multiple comparisons of repellency, avoidance, and average mortality values between the test materials.

## 3. Results

The mean repellency, mean avoidance, and mean mortality values obtained from adult female *P. papatasi* for the essential oils and other test compounds are listed in order of static air repellent efficacy in Table 1. The results of ANOVA indicate significant differences among the tested materials in regards to repellency (*F*_12, 29_ = 9.758, *p* < 0.0001), avoidance (*F*_12, 29_ = 22.491, *p* < 0.0001), and toxicity/mortality (*F*_12, 29_ = 3.967, *p* < 0.0012) against sand flies. All test materials, except for nonanoic acid, demonstrated significantly higher repellency (*p* < 0.05) than did the solvent only control against sand flies. Catnip oil exhibited the highest (100%) repellency, and nonanoic acid showed the lowest (30.8%) repellency. Three test materials (catnip oil, amyris oil, and undecanoic acid) achieved 100% sand fly avoidance. While all test materials achieved significantly higher sand fly avoidance than the solvent only control (*p* < 0.05), no significant difference (*p* > 0.05) was found among the twelve test materials. While nonanoic acid was the least repellent to sand flies, it was as effective as other test materials in causing sand flies to avoid the treated filter papers. In addition, nonanoic acid caused the highest sand fly mortality (53.3%), followed by undecanoic acid (16.7%) and the essential oil of *Lippia graveolens* (15.0%). Mortality caused by nonanoic acid was significantly higher (*p* < 0.05) than that of *Lippia graveolens* oil, but it was not statistically different from that of undecanoic acid (*p* > 0.05).

Different patterns regarding the time–repellency response of sand flies to the test compounds were noted when the mean repellency rate was plotted for each observation time (Figure 2). Several compounds showed increasing repellency over time (Figure 2A), whereas other compounds exhibited relatively stable repellency values over time (Figure 2B). Still other compounds generally exhibited decreasing repellent activity over increasing time (Figure 2C). The constant spatial repellency of *L. graveolens* essential oil for the first 60 min was followed by a slowly decreasing repellency over the next 120 min (Figure 2C). An initial increase in repellency by nonanoic acid during the first 30 min of exposure was rapidly lost thereafter. *L. graveolens* oil, undecanoic acid, and nonanoic acid produced elevated fly mortality compared to the negative (acetone only) control (Table 1).

The mean repellency and avoidance values for the essential oils of *N. cataria* and *A. balsamifera* were further compared to those of DEET and IR3535 at four different concentrations (1%, 0.5%, 0.25%, and 0.125%) (Figure 3). The results of ANOVA indicate significant differences among the four test materials in regards to sand fly repellency at the test concentrations of 1% (*F*_3, 9_ = 18.400, *P* = 0.0004) and 0.5% (*F*_3, 6_ = 37.710, *p* = 0.0003), but not at 0.25% (*F*_3, 7_ = 3.769, *p* = 0.067). Significant differences among the four test materials were found in regards to sand fly avoidance at 0.5% (*F*_3, 9_ = 13.494, *p* = 0.0045), but not at 1% (*F*_3, 9_ = 1.573, *p* = 0.263) or 0.25% (*F*_3, 6_ = 4.302, *p* = 0.061). *N. cataria* and *A. balsamifera* oils exhibited generally higher repellency and avoidance than did DEET and IR3535 at concentrations of 1%, 0.5%, and 0.25%, although at a significant level (*p* < 0.05) only at 0.5% for repellency. There also appeared to be dose-dependent repellent and avoidance responses for these four repellent materials. DEET generally performed better than IR3535, but IR3535 resulted in a significantly higher (*p* < 0.05) avoidance than DEET at the concentration of 0.125%.

The results of enzyme inhibition assays (Table 2) show that none of the inhibitors significantly inhibited the recombinant AChE of *P. papatasi*. Nevertheless, the exposure of sand flies to decreasing concentrations of some of the materials revealed threshold levels at which the measured spatial repellency dropped below 50%. Table 3 lists the threshold concentrations of the test materials required for 50% sand fly repellency. These threshold values demonstrated higher relative spatial repellent activity for *N. cataria*, *A. balsamifera*, and *M. piperita* oils compared to those of DEET and IR3535. As compared to IR3535, DEET exhibited a two-fold higher relative sand fly repellent activity at a low concentration.

## 4. Discussion

Essential oils and other natural compounds have been extensively evaluated against mosquitoes and other insect pests [65], but much less so for sand flies. Our study was the first one to utilize the static air repellency bioassay technique for the evaluation of repellents against sand flies. Four of the essential oils (catnip, amyris, tea tree, and peppermint oils) were found to show higher sand fly repellency against adult female sand flies than either DEET or IR3535. However, the results of the statistical analysis did not show significant differences among the materials in regards to repellency, with the exception of nonanoic acid, which was significantly less repellent than the top four essential oils. Sand flies showed 100% avoidance to filter paper at the end of the test tubes treated with catnip oil, amyris oil, and undecanoic acid. Overall, sand flies showed a high contact avoidance (>80.5%) of the test materials, regardless of the level of repellency measured in this study.

The biting deterrence of IR3535 and DEET for *P. mascitti* and *P. duboscqi* was previously reported [43]; however, spatial or contact repellency was not determined. Spatial repellency and avoidance are both important measures that contribute to biting deterrence in mosquitoes [63]. Nonanoic acid and undecanoic acid were the only materials that resulted in significant sand fly mortality. *L. graveolens* oil also appeared to produce elevated mortality, but this result was not statistically different from that of the control. Nonanoic acid in plant products killed the larvae of *Cx. pipiens* [66]. Undecanoic acid was not insecticidal to *An. gambiae* [67]. The finding that the exposure of sand flies to the least repellent nonanoic acid resulted in the highest mortality (53.3%) suggests that nonanoic acid may possess high fumigant toxicity against adult sand flies, identifying it as a potential fumigant insecticide to control adult sand flies, particularly in more closed environments, such as for use against newly emerged adult sand flies in rodent burrows.

Our primary goal was to compare the sand fly repellency of the ten essential oils and natural compounds with that of DEET and IR3535, the active ingredients used in commercial contact insect repellent products. We followed the previously published protocol, and no screens were used to prevent sand flies from making contact with the treated filter paper. The static air repellency bioassay allowed for the measurement of both the contact and spatial components of sand fly repellency (Table 1, Figure 2 and Figure 3). We realized that any contact with the treated filter paper could affect sand fly behavior. Therefore, the spatial repellency we measured and presented in this report may not be exclusively due to spatial repellency. Future specific studies regarding spatial repellency using the static air bioassay technique would require the addition of screens that separate the flies from the treated filter paper at both ends of the glass tube and the use of the most potent spatial repellent compounds, such as metofluthrin or transfluthrin, as the positive control.

Comparison of the relative spatial repellent activity for each of the test materials over the course of time (Figure 2) revealed that DEET, *P. dioica* oil, tea tree oil, decanoic acid, undecanoic acid, and IR3535 showed a general trend of increasing spatial repellency as time progressed. This observation suggests that a gradual accumulation of volatile repellent components occurred within the static air chamber [16,63]. In contrast, catnip oil, amyris oil, peppermint oil, and undecanoic acid exhibited relatively constant spatial repellency over time. These materials apparently established their effective spatial repellent gradients rapidly within the chamber, while *L. graveolens* oil and nonanoic acid tended to show decreasing spatial repellency over time. The repellent volatile components of *L. graveolens* oil and nonanoic acid may be unstable or rapidly depleted, or perhaps the behavioral response of *P. papatasi* may change due to habituation or toxic effects, as evidenced by the mortality results listed in Table 1.

The observed *P. papatasi* repellency produced by the saturated fatty acids (decanoic acid (C10) > undecanoic acid (C11) > dodecanoic acid (C12) > nonanoic acid (C9)) is in general agreement with that reported for *Ae. aegypti* [18,36,37]. This suggests that the repellent activity against *P. papatasi* was related to fatty acid chain length, C10 > C11 > C12 >> C9. If that is the case, the spatial repellency to sand flies in our system likely reflects relative volatility, as asserted by Paluch et al. [16], and decreasing volatility from C10–C12, with differential insect sensitivity for C9, which exhibited significant mortality to the nonanoic acids, likely preventing the flies from moving away from the nonanoic acid. In contrast, the relative sand fly avoidance (contact repellency) of these saturated fatty acids was very high for C9–C11 (≥95%), but somewhat reduced for C12 (91%), suggesting a slightly reduced contact sensitivity for dodecanoic acid. The sand fly repellency over time (Figure 2) was similar for decanoic acid (C10) and undecanoic acid (C11), gradually increasing over time, while dodecanoic acid (C12) appeared to produce relatively stable repellency over time, in contrast to nonanoic acid (C9), which produced decreasing repellency over time. All of the saturated fatty acids at 1% concentration exhibited lower repellency to sand flies than that of DEET.

As shown in Figure 2B, *N. cataria* and *A. balsamifera* oils exhibited sand fly repellency values very close to 100% throughout the 180 min time period, clearly exceeding the performance of all of the other materials tested at 1% concentration, similar to previous results reported by Paluch et al. [63]. The superior performance of *N. cataria* and *A. balsamifera* oils was further evidenced (Figure 3) by their relative repellency and avoidance activities when compared to those of the synthetic repellents, DEET and IR3535, at reduced concentrations. *N. cataria* oil, *A. balsamifera* oil, and DEET exhibited dose-dependent repellent and avoidance values. IR3535 exhibited reduced repellent and avoidance values when its concentration was decreased from 1% to 0.5%, but the repellent and avoidance values increased when sand flies were exposed to even lower concentrations ranging from 0.5% to 0.125%. This observation suggests a possible behavioral interference of IR3535 at the higher concentrations tested. The threshold concentrations reducing sand fly repellent activity below 50% for *N. cataria, A. balsamifera*, and *M. piperita* oils were two-, four-, and eight-fold lower, respectively, compared to those of DEET (Table 3).

In our study, we observed that the sand flies exposed to IR3535 exhibited erratic or disoriented behavior, and as shown in Table 1, also exhibited elevated mortality, suggesting IR3535 toxicity. Weeks et al. [7] reported that *P. papatasi* sand flies were observed to approach human skin treated with IR3535, but to suddenly veer away, suggesting a limited spatial component of repellency. Shrestha et al. [42] reported that bitter-sensing gustatory receptor neurons were essential for IR3535 detection, and that together, DEET and IR3535 exerted synergistic effects. Low concentrations of IR3535 act on the muscarinic acetylcholine receptors of the insect neurosecretory cells, inducing increased intracellular calcium, resulting in an increased sensitivity of the nicotinic acetylcholine receptors to thiacloprid [42]. Some terpene constituents of essential oils have been shown to inhibit arthropod acetylcholinesterase [44,68]. Our observation of erratic behavior and mortality, together with the report of Moreau et al. [41] regarding the acetylcholine receptors and intracellular calcium, suggested that IR3535 might exert direct effects on cholinergic targets such as acetylcholinesterase. The elevated mortality resulting from nonanoic acid and undecanoic acid also suggested the potential involvement of AChE, an essential neural enzyme and the target of organophosphate and carbamate pesticides, prompting us to test these individual chemicals along with IR3535 and DEET for inhibition of the recombinant acetylcholinesterase of *P. papatasi* [54,55]. However, in our experiments, nonanoic acid, undecanoic acid, IR3535, and DEET were found to exhibit little or no inhibition of *P. papatasi* acetylcholinesterase (Table 2).

*A. balsamifera* essential oil is comprised of valerianol, eudesmol, elemol, guaiol, agarospirol, hedycaryol, and other sesquiterpenes [69,70,71]. *A. balsamifera* oil and elemol exhibited high levels of contact repellency (avoidance) but low spatial repellency to *Ae. aegypti*, which contrasted with the opposite trend for *N. cataria* oil against the same mosquito species [63]. However, a mixture of *A. balsamifera* and *N. cataria* oils produced high spatial and contact repellency to *Ae. aegypti* mosquitoes [63]. Schultz et al. [65] reported that elemol exhibited similar residual spatial repellent activity to *Cx. pipiens* to that of DEET at 0.1% application after 180 min.

*M. piperita* oil is composed mostly of monoterpenes, including menthol, menthone, menthofuran, menthyl acetate, 1,8-cineole, and cis-carane [72,73,74]. This essential oil exerted 30 min of protection from bites of *Ae. aegypti* when applied to human skin at 25–50% concentration, and 45 min of protection when applied at 100% concentration, compared to 5–6 h of protection by 25% DEET [75]. The application of 0.1 mL *M. piperita* oil to a 25 cm^2^ area of human forearm provided 100% protection from blood-starved *Ae. aegypti* bites for 150 min, with a progressive decrease in protection thereafter [76]. Herein, sand flies exhibited a higher avoidance (contact repellency) than spatial repellency for peppermint oil (Table 1), which slowly increased over time (Figure 2), and exhibited 50% repellency at an eight-fold lower concentration than that of DEET (Table 2). Our data suggest that several constituents of *M. piperita* oil contribute to sand fly repellency, in which volatile components like menthol contribute to higher spatial repellency compared to that of DEET at a low concentration, as well as less volatile elements contributing to high contact repellency (avoidance) and the durability and slow increase in spatial repellency.

*N. cataria* oil is composed of nepetalactones, with small amounts of α-pinene, geraniol, β-caryophyllene, and other minor components [77]. The relative yield and composition of the oil varies depending on the plant parts collected, the growth stage, and the time and manner of oil extraction, with generally the highest yield and nepetalactone content in the flowering stage [78,79,80,81,82]. Thus, the actual composition of *N. cataria* oil can vary depending on multiple factors. Reichert et al. [83] reported a new cultivar of *N. cataria* that exhibited improved growth character and volatile oil production, with strong repellency to a wide range of vector and pest species that should be more amenable to commercial production and harvesting than previous stocks. A comparison of DEET with the high nepetalactone *N. cataria* oil or purified nepetalactone demonstrated that they exhibited equivalent inhibition to the landing of *Ae. aegypti* at 0.1% to 1.0% application rates, and that the *N. cataria* oil or purified nepetalactone was more effective at inhibiting *Ae. aegypti* landing than DEET at 0.01%. Similarly, we observed a higher relative repellency against *P. papatasi* of *N. cataria* oil compared to that of DEET at low concentrations (Figure 3).

A comparison of 10% DEET with *N. cataria* oil or purified nepetalactone revealed that DEET retained its high repellent activity longer than the nepetalactone or *N. cataria* oil [84]. Simmons et al. [85] reported that the vapor pressures exerted by *N. cataria* oil and its two major nepetalactone isomers are similar to that of DEET. In contrast, Paluch et al. [63] reported that the calculated vapor pressures for the two major nepetalactone isomers of *N. cataria* oil were significantly higher (1.75 mmHg) than that of DEET (0.58 mmHg). The higher relative repellency of *N. cataria* oil at reduced concentrations that we and other researchers [16,63,84] observed and the extended repellency of DEET compared to that of nepetalactones and *N. cataria oil* over time are evidence suggesting that DEET is less volatile than the nepetalactones. Under this scenario, the higher volatility of the nepetalactones would result in a higher repellent activity at a low concentration compared to that of DEET but would also become depleted more rapidly over time.

The chemical modification or advanced formulation of nepetalactones may reduce their volatility, thus providing the potential for longer lasting repellent activity [86]. The repellency of *N. cataria* oil to *Ae. aegypti* and *D. melanogaster* is mediated by TRPA1, a widely conserved chemical irritant receptor [87]. Insect TRPA1 mutants may no longer be repelled by *N. cataria* oil or nepetalactone. *N. cataria* oil does not activate human TRPA1, which provides evidence that this essential oil, or nepetalactone, is insect selective. Therefore, insect TRPA1 can be targeted for the development of novel, safe insect repellents [88]. Studies on *N. cataria* oil components and mixtures of sesquiterpenes or essential oils indicated that individual oils or constituent components of essential oils exhibit significantly increased repellent activity or duration of repellent efficacy in several pest species, suggesting potential synergistic effects [16,63,84,85,89,90].

Essential oils and their components exhibit various repellent and insecticidal activities to different insect pests and disease vector species. The static air repellency bioassay used in this study and other in vitro repellency assays generally measure insect responses to test materials in the absence of host attractant cues. This is a simpler approach to elucidate insect responses to interaction with specific chemicals (mode of action studies) than in vivo systems measuring biting deterrence, which add the complexity of interactions with various host chemical and behavioral cues that influence insect responses [91]. Measurement of in vitro repellency is also useful to assess the potential of compounds to protect premises [65]. Further research is necessary to establish modes of action of the natural products tested here against *P. papatasi* and to elucidate potential synergistic interactions between known repellent chemicals and natural chemistries developed for sand fly control.

## Figures and Tables

**Figure 1 insects-15-00155-f001:**
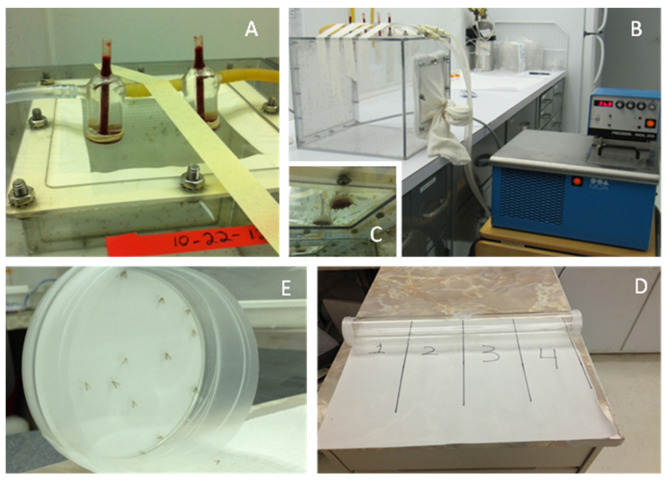
(**A**,**B**) adult Plexiglass cage with blood feeding tubes at top; (**C**) female sand flies feeding on blood; (**D**) static air repellency apparatus indicating divisions into quadrants; (**E**) adult female sand flies on the control filter at end of the repellency tube.

**Figure 2 insects-15-00155-f002:**
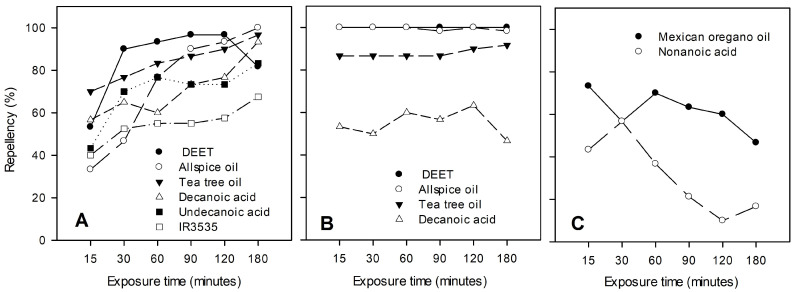
Mean repellencies to female *P. papatasi* flies vs. time for exposure at 1% dilution of test material (157.2 µg/cm^2^). (**A**) Test materials exhibiting increasing repellency over time; (**B**) test materials exhibiting relatively constant repellency over time; (**C**) test materials exhibiting decreasing repellency over time.

**Figure 3 insects-15-00155-f003:**
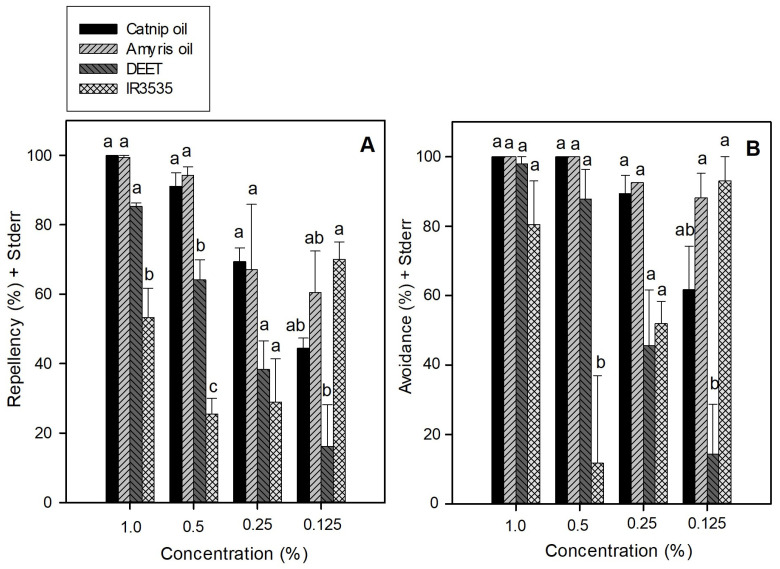
(**A**) Mean repellency of test compounds at decreasing concentrations. (**B**) Mean avoidance values produced by test compounds at decreasing concentrations: 1% = 157.2 µg/cm^2^; 0.5% = 78.6 µg/cm^2^; 0.25% = 39.3 µg/cm^2^; 0.125% = 19.65 µg/cm^2^. Means in each test concentration that have different letter(s) are significantly different (Tukey–Kramer HSD test, *p* < 0.05).

**Table 1 insects-15-00155-t001:** Mean values of adult female *P. papatasi* repellency, avoidance, and cumulative mortality exhibited by test material exposure at 1% dilution (157.2 µg/cm^2^) in a static air olfactometry.

	Repellency (%)		Avoidance (%)		Mortality (%)	
Chemical (1%)	Mean (Stderr)		Mean (Stderr)		Mean (Stderr)	
Catnip (*Nepeta cataria*) oil	100.0 (0.0)	a	100.0 (0.0)	a	0	b
Amyris (*Amyris balsamifera*) oil	99.4 (0.6)	a	100.0 (0.0)	a	0	b
Century tea tree (*Melaleuca alternifolia*) oil	88.9 (4.4)	a	99.3 (0.7)	a	5.0 (2.9)	b
Peppermint (*Mentha piperita*) oil	88.3 (4.4)	a	98.1 (1.9)	a	5.0 (2.9)	b
DEET	85.3 (1.0)	ab	98.0 (2.0)	a	1.7 (1.7)	b
Allspice (*Pimenta dioica*) oil	73.3 (1.7)	ab	93.1 (3.8)	a	3.3 (3.3)	b
Decanoic acid	70.8 (8.5)	ab	94.6 (4.0)	a	0	b
Undecanoic acid	70.0 (3.5)	ab	100 (0.0)	a	16.7 (6.0)	ab
Mexican oregano (*Lippia graveolens*) oil	61.7 (17.8)	ab	90.2 (2.8)	a	15.0 (10.4)	b
Dodecanoic acid	55.0 (13.5)	ab	90.9 (3.1)	a	0	b
IR3535	54.6 (8.9)	ab	80.5 (12.5)	a	7.5 (4.3)	b
Nonanoic acid	30.8 (28.1)	bc	98.6 (1.4)	a	53.3 (24.2)	a
Solvent (acetone) only	−10.7 (9.1)	c	−15.8 (13.7)	b	0	b

Tests for each chemical included repellency at 1% test material exposure (averaged over 180 min) for three replicate experiments (n = 3), except for IR3535 (n = 4) and the solvent control (n = 5). The means in each column that have different letter(s) are significantly different (Tukey–Kramer HSD test, *p* < 0.05).

**Table 2 insects-15-00155-t002:** Results of enzyme (rPpAChE) inhibition assays.

Inhibitor Concentration (µM)	% Residual rPpAChE1 Activity (Compared to No Inhibitor) ± Std. Dev. ^*^			
	Nonanoic Acid	Undecanoic Acid	IR3535	DEET
0.0508	99.7 ± 1.8	101.6 ± 1.2	106.9 ± 1.2	97.6 ± 0.6
0.152	102.1 ± 0.7	100.8 ± 0.4	107.2 ± 2.2	96.6 ± 0.4
0.457	102.9 ± 1.1	102.7 ± 0.5	107.5 ± 0.9	97.5 ± 0.7
1.37	103.7 ± 0.5	101.9 ± 1.1	100.0 ± 7.0	94.5 ± 0.5
4.12	103.3 ± 0.8	102.6 ±0.3	105.0 ± 0.4	96.8 ± 0.7
12.3	103.7 ± 1.5	102.5 ± 0.5	104.5 ± 0.6	95.1 ± 1.8
37	102.2 ± 0.5	101.9 ± 0.9	106.4 ± 2.1	94.7 ± 1.3
111	101 ± 1.7	101.3 ± 0.7	105.7 ± 4.0	90.8 ± 3.0
333	98.5 ± 1.5	100.3 ± 1.4	103.0 ± 3.9	77.1 ± 13.1
1000	96.0 ± 2.3	98.2 ± 0.4	103.8 ± 5.0	93.2 ± 1.4

^*^ Means of three replicates.

**Table 3 insects-15-00155-t003:** Threshold concentrations of test materials required for 50% sand fly repellency.

Material	Concentration at Which Spatial Repellency Fell Below 50%
Catnip (*Nepeta cataria*) oil	0.125% (11.65 µg/cm^2^)
Amyris (*Amyris balsamifera*) oil	0.06125% (9.825 µg/cm^2^)
Century tea tree (*Melaleuca alternifolia*) oil	ND *
Peppermint (*Mentha piperita*) oil	0.03125% (4.9125 µg/cm^2)^
DEET	0.25% (39.31 µg/cm^2^)
Allspice (*Pimenta dioica*) oil	ND
Decanoic acid (C10:0)	ND
Undecanoic acid (C11:0)	ND
Mexican oregano (*Lippia graveolens*) oil	ND
Dodecanoic acid (C12:0)	ND
IR3535	0.5% (78.62 µg/cm^2^)
Nonanoic acid (C9:0)	1%
Solvent (acetone) only control	ND

* ND = not done.

## Data Availability

Analyzed data from this study are presented within the manuscript. Original data are available from the corresponding author upon request.

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
