# Peer review of "In Vitro Evaluation of Essential Oils and Saturated Fatty Acids for Repellency against the Old-World Sand Fly, Phlebotomus papatasi (Scopoli) (Diptera: Psychodidae)"

_insects, 2024, doi:10.3390/insects15030155_

Round 1

Reviewer 1 Report

Comments and Suggestions for Authors

Overall comments: This is an overall well-written study evaluating the repellency effects of some essential oils and saturated fatty acids against Ph. papatasi using an in vitro bioassay. The study takes full points for investigating new tools for the control of this highly important yet neglected group of insects. The authors have sufficiently explained the rational and the design of their research, showcasing new findings which are certainly worthy of publication. I do, however, have two important concerns that the authors should address prior to publication involving a) the design of the bioassay for assessing spatial repellency and b) the products used as the gold standard (DEET, IR3535). Regarding concern a) The described bioassay allows insects to come in direct contact with the active ingredient therefore it does not assess the pure spatial repellency effect (but a combination of spatial and contact repellency). Could the authors comment on this? Even a small amount of active ingredient coming in direct contact with the flying insects could significantly alter their behavior. Were the authors observing the insects for the duration of the experimentation to ensure that for the spatial repellency values insects had not touched the filter paper?  If this cannot be confirmed, then I suggest that the data (repellency + avoidance) are combined, and the active ingredients are evaluated based on overall repellency. Regarding concern b) Volatile pyrethroids are currently the most effective spatial repellents, and among the most studied active ingredients are the following: metofluthrin, transfluthrin, allethrin, prallethrin. On the other hand, DEET and IR3535 are best known for their contact repellency effect. They have low volatility and as such can remain on treated surfaces without volatilizing into the surrounding atmosphere in a significant degreeIf the authors wanted to assess “spatial” repellency effect of the new products why not comparing them against the most effective, highly volatile compounds mentioned above? The authors should make it clear during the discussion when comparisons are made between the new products tested and DEET/IR3535 that the latter two are best known for their contact/topical repellency and not for their spatial repellency.

Specific comments/corrections

Lines 139-151: If there is a rearing protocol available online perhaps it would be best to cite the protocol instead of describing the rearing process. 

Line 172: Why were the material allowed to dry for an hour? It seems there would be a risk of losing some of the most highly volatile components. Could the authors explain?

Lines 185-191: There are no results provided on the inhibition of sand flies on acetylcholinesterase. The authors should either provide the results or remove this section entirely. 

Line 192: Could the authors provide reference/s for the formulas used?

Line 215: Please correct 00% to 100%. Double check all numbers to ensure all are accurate. Also make sure you reference in text at least once all tables and figures.

Lines 231-241:  “Type-1, 2, 3” are mentioned for the first time here. Either definition should be provided earlier or this should be removed entirely. 

Line 261: Why are the mean avoidance values not presented here (similarly to the next figure)? 

Lines 281-284: Please, remove.

Sand flies are true Dipterans and as such should not be written as one word. Please, correct “sandfly” to “sand fly” across the text. Also, correct nonanoate to nonanoic acid when appropriate across the text. 

Author Response

Please see the Word file attached.

Reviewer 2 Report

Comments and Suggestions for Authors

The manuscript "In vitro evaluation of essential oils and saturated fatty acids for repellency against the old-world sandfly, Phlebotomus papatasi (Scopoli) (Diptera: Psychodidae)" by Kevin B. Temeyer and colleagues investigates the repellent activities of six essential oils and four saturated fatty acids against adult sandflies and the potential activity of DEET and IR3535 against acetylcholinesterase in the same insect. The objectives of this study are interesting and original. In my opinion the results obtained are hardly sufficient to be published in this journal. Anyway the authors clearly described the methodologies and results of their experiments, although in materials and methods it would be better to include higher quality of the photos of figure 1, in particular better picture  (taken more closely) of static air repellency apparatus during the experiments (with the insects inside). About figure 2, I suggest moving it after the line n. 245.

References regarding the inhibition of AChE in sandfly should be added in the introduction to better describe this topic. Add more references regarding advantages of natural products such as essential oils compared to synthetic repellents.

Delete the lines n. 281, 282, 283 and 284.

Describe the reference [40] more fully (e.g. the experiments were on larvae, adults, cell lines etc...).

Line 28: against

Line 57: [1,2].  The old-world

Line 60: [3-7].  Adult female

Line 127: U.S. Department of Agriculture-­­ Agricultural

Line 166: DDEET

Line 218: materials.  While

Line 234: (n = 5).  Means

Line 241: L. graveolens italics

Line 244: L. graveolens italics

Author Response

Please see the Word file attached.

Reviewer 3 Report

Comments and Suggestions for Authors

In this study, the authors test a variety of essential oils and fatty acids against the sandfly Phlebotomus papatasi, an important vector of Leishmania and other pathogens. There is an increasing research effort to identify organic compounds that may be effective repellents and insecticides against arthropod vectors, but tests against sandflies are not very common in the literature, so this article makes welcome progress in this field.

This study identified a number of essential oils that showed good repellency against P. papatasi, with favourable results compared to DEET, that should be developed further, and as authors say in the Discussion should be researched further to identify modes of action and synergistic activities with other chemicals. Most of the candidates have also been shown to be good repellents against various mosquito species, so may make good general insect repellents.

Overall this is a good contribution to the field and is well-written and organised. My major concern with this article is that certain experiments (acetylcholinesterase inhibition assays) are described in the Introduction, Methods and even Discussion, but the findings from these are omitted from the Results.

Specific comments:

The Introduction is well written and provides a good background, with the impact and aims of the study clearly shown. However, there is some description (lines 111-114, and 118-120) of experiments for which results are not presented in the paper.

Methods
Figure 1 - some of the images need to be higher resolution as they are not very clear, especially C and E.

Section 2.4 - Methods are described for an assay to measure acetylcholinesterase inhibition, but no results are presented for this.

Results
Results are mostly clearly described but there are some editing needed to improve clarity and presentation: 
line 215: should say 100% rather than 00%.

line 224: "It" should specify 'Undecanoic acid and L. glaveolens oil'.

line 226-7: This line would be better placed at line 218 (after sentence ending "...twelve test materials.") in the section describing avoidance.

line 244-5: Suggest delete this sentence, as it is repetitive of lines 220-225. Plus two of these materials were not significant compared to control, so it is also misleading.

Figure 2 - for part A, there is a label covering some of the graph "Document was last saved: just now" which needs to be removed.

lines 281-284 - remove this template text.

Discussion
Lines 364-366 refer to Results that are not presented in this paper.

Author Response

Please see the Word file attached.

Round 2

Reviewer 3 Report

Comments and Suggestions for Authors

I am satisfied that the authors have addressed all concerns raised by reviewers, and recommend that the paper be accepted for publication.